# VAENAS: Sampling Matters in Neural Architecture Search

## Abstract

Neural Architecture Search (NAS) aims at automatically finding neural network architectures within an enormous designed search space. The search space usually contains billions of network architectures which causes extremely expensive computing costs in searching for the best-performing architecture. One-shot and gradient-based NAS approaches have recently shown to achieve superior results on various computer vision tasks such as image recognition. With the weight sharing mechanism, these methods lead to efficient model search. Despite their success, however, current sampling methods are either fixed or hand-crafted and thus ineffective. In this paper, we propose a learnable sampling module based on variational auto-encoder (VAE) for neural architecture search (NAS), named as *VAENAS*, which can be easily embedded into existing weight sharing NAS framework, e.g., one-shot approach and gradient-based approach, and significantly improve the performance of searching results. VAENAS generates a series of competitive results on CIFAR-10 and ImageNet in NasNet-like search space. Moreover, combined with one-shot approach, our method achieves a new state-of-the-art result for ImageNet classification model under 400M FLOPs with 77.4% in ShuffleNet-like search space. Finally, we conduct a thorough analysis of VAENAS on NAS-bench-101 dataset, which demonstrates the effectiveness of our proposed methods.

## 1 Introduction

Deep neural networks have greatly pushed the frontier of various influential applications by designing novel neural architectures (Krizhevsky et al. (2012); Goodfellow et al. (2014); He et al. (2016)). Automatic model design of neural network architectures without human intervention, known as *neural architecture search (NAS)*, has drawn much attention of the community recently. It has resulted in state-of-the-art performance in the domain of image recognition (Zoph et al. (2018); Real et al. (2019)), object detection (Ghiasi et al. (2019); Chen et al. (2019)) and semantic segmentation (Liu et al. (2019)).

Generally, the magnitude of search space for NAS tasks is enormous. For example, NasNet (Zoph et al., 2018) presents a search space with $6 \times 10^9$ possible cells. Searching on such huge designed space cost 2400 GPU days. Weight sharing mechanism has shown to be a promising avenue for efficient NAS. Latest algorithms on efficient NAS fall into two categories: one-shot approaches (Bender et al. (2018)) and gradient-based approaches (Liu et al. (2018b)). In one-shot approaches, prior works focus on adopting a fixed sampling strategy (Guo et al. (2019); Bender et al. (2018); Chu et al. (2019)). In gradient-based approaches, search is typically performed without sampling procedure (Liu et al. (2018b); Xie et al. (2018)) or hand-crafted sampling (Liu et al. (2018a)). Despite the success of these NAS methods on various benchmarks, however, these sampling approaches do not interactively learn the architecture distribution as the search process goes along, which makes the sampling procedure ineffective.

In this paper, we propose a learnable and interactive architecture sampling module based on VAE for NAS, named as *VAENAS*. There are two advantages of VAENAS: 1) it learns the good-performing architecture distribution which could be used to reduce search space. 2) it can be embedded into existing NAS framework and improve the performance of current NAS methods.

After developing the generic VAENAS approach, we study in detail the application of VAENAS module via two mainstream NAS approaches: one-shot approach (Brock et al. (2017); Guo et al.

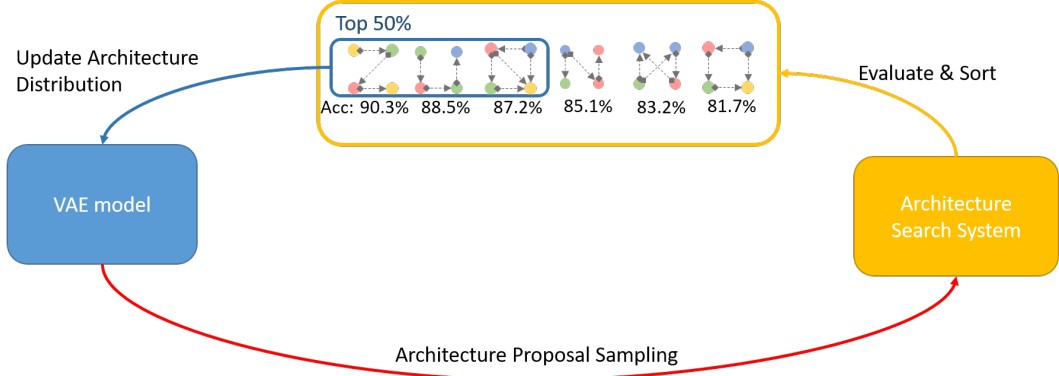

Figure 1: The illustration of VAENAS mechanism. VAENAS mechanism contains two modules, one is VAE model, and the other is an architecture search system. These two modules are updated alternately. Specifically, three steps are carried out on VAENAS mechanism. 1) Architectures are sampled from VAE model and fed into the architecture search system. 2) The architecture search system evaluates and sorts sampled architectures. 3) The VAE model is trained with top-50% architectures.

(2019)) and gradient-based approach (Liu et al. (2018b); Cai et al. (2018)), both of which have obtained state-of-the-art performance on neural architecture search tasks. We show how VAENAS module can be embedded in both approaches and make performance improvement consistently.

We validate VAENAS with various search space on two benchmark datasets (CIFAR-10 and ImageNet) for image recognition. On NASNET-like search space, we apply VAENAS on both gradient-based methods and one-shot methods. Specifically, combined with one-shot approach, VAENAS achieves 2.26% test error on CIFAR-10 and 75.8% accuracy on ImageNet, outperforming state-of-the-art NAS methods. On a Shufflenet-like search space, VAENAS combined with one-shot achieves 77.4% top-1 accuracy with 365M FLOPs on ImageNet classification, outperforming state-of-the-art Efficient-B0 by 1.1% with 6.5% less computational complexity. Finally, we perform a thorough analysis of VAENAS on NAS-101 benchmark, to show the effectiveness of our proposed architecture sampling methods.

## 2 RELATED WORK

**Neural Architecture Search.** Recently the design of efficient neural networks has largely shifted from leveraging human knowledge to automatic methods, which is known as neural architecture search(NAS). Early NAS methods adopt reinforcement learning (RL)(Zoph et al., 2018) or evolutionary strategy (Real et al., 2019) to search on thousands of individually evaluated networks, which is computational-consuming. Recent works focus on efficient search methods, which falls into two categories: one-shot approach (Guo et al. (2019); Chu et al. (2019)) and gradient-based approach (Liu et al. (2018b); Cai et al. (2018)), both achieve state-of-the-art results on a series of benchmark dataset (Chen et al. (2019); Ghiasi et al. (2019); Liu et al. (2019)) with various search space. We argue that sampling method matters in NAS and a learnable sampling algorithm can significantly improve the NAS performance.

**Sampling.** Some prior works have applied sampling methods to NAS framework. For one-shot approach, Bender et al. (2018) randomly zero out a subset of the operations during super-network training. Guo et al. (2019); Stamoulis et al. (2019) adopt uniform sampling and Chu et al. (2019) employ fair uniform sampling to reduce training bias in one-shot models. For gradient-based approach, Cai et al. (2018) propose a multinomial distribution sampling to alleviate large memory consumption issue in conventional gradient NAS methods (Liu et al. (2018b)). Our approach presents a new perspective of sampling methods of NAS application.

---

**Algorithm 1** VAE Training Pipeline

---

**Input:** Architectures and corresponding performance set $S = \{(\alpha_i, ACC_i)\}_i^N$, # of training itera-
    tions $n_{iters}$, parameters of VAE encoder $\phi$, parameters of VAE deocder $\theta$, ranking ratio $K$.
**Output:** Updated VAE parameters $\phi, \theta$
 1: Rank $S$ by the descending order of $ACC_i$;
 2: Select top $K\%$ architectures as VAE training dataset $S_K$, $S_K \subset S$ ;
 3: **for** $j = 1, ..., n_{iters}$ **do**
 4:     Optimize $\phi$ and $\theta$ with $S_K$.
 5: **end for**

---

## 3 VAENAS

### 3.1 MOTIVATION

Many prior works in NAS have discussed modeling the architecture distribution $\Gamma(\alpha)$. The typical $\Gamma(\alpha)$ is to adopt either fixed architecture distribution (e.g., uniform distribution Bender et al. (2018); Guo et al. (2019)), or hand-crafted architecture distribution (e.g., Gumbel softmax distribution in Liu et al. (2018b); Xie et al. (2018)).

Conceptually, it is expected that the architecture distribution $\Gamma(\alpha)$ can interactively learn from the weight sharing super-network as the searching process goes along. To this end, we propose a learnable architecture distribution, where the distribution $\Gamma_t(\alpha)$ at time $t$ is learned from the new incoming architecture $\alpha$ and prior architecture distribution at $t-1$ time is $\Gamma_{t-1}(\alpha)$, we can formalize the distribution learning process as:

$$\Gamma_t(\alpha) \leftarrow \mathcal{T}(\Gamma_{t-1}(\alpha), Metric_{t-1}(\alpha)) \tag{1}$$

where $Metric$ denotes the predefined metric, e.g., accuracy or loss, and $\mathcal{T}$ denotes the stochastic process. In equation 1, the distribution of architectures at time $t$ is learned by both the prior knowledge on architecture distribution $\Gamma(\alpha)$ and new architectures data $\alpha$ at $t-1$ time, as in Figure 1. As the architecture search proceeds, we randomly sample multiple architectures, rank these architectures according to the predefined metric. The $\mathcal{T}$ incrementally refines the distribution of architectures by continually learning the good-performing architectures from the weight sharing network. Alternatively, the weight sharing network updates the architectures weights, where the architectures is sampled from $\mathcal{T}$. There are many methods that can be used to model this distribution, and we choose VAE due to its training stability as well as the convenience to be embedded into the existing NAS frameworks.

### 3.2 THE USAGE OF VAE

VAE approximates the marginal likelihood of architecture distribution $\log p_\theta(\alpha)$ by maximizing variational lower bound:

$$-D_{KL}(q_\phi(z|\alpha)||p_\theta(z)) + E_{z \sim q_\phi(z|\alpha)}[\log p_\theta(\alpha|z)] \tag{2}$$

Specifically, the left term is the Kullback-Leibler divergence between the approximated posterior (recognition model) $q_\phi(z|\alpha)$ and the prior $p(z)$. And the right term is the expectation of reconstruction loss where $p_\theta(\alpha|z)$ denotes variational inference. As a result, to sample $\alpha$ from the marginal distribution $p_\theta(\alpha)$, VAE samples latent variable $z$ according to the prior distribution $p_\theta(z)$ and feed $z$ into the inference model $p_\theta(\alpha|z)$. Commonly, the prior distribution $p_\theta(z)$ can be set as standard Gaussian distribution $N(0, I)$.

Let $\boldsymbol{\alpha} = \{o_i\}_i^n$ be a sequence of operation $o_i$ representing an architecture string, where $n$ is the total number of nodes in a network or cell. We use an encoder $E$ to take $\boldsymbol{\alpha}$ as input and maps it into parameters $\boldsymbol{\mu}_\alpha$ and $\boldsymbol{\sigma}_\alpha$ corresponding to a normal distribution $N(\boldsymbol{\mu}_\alpha, \boldsymbol{\sigma}_\alpha)$. Symmetrically, a decoder $D$ is used to map $\boldsymbol{z}_e$ to a reconstructed architecture $\bar{\boldsymbol{\alpha}}$, where $\boldsymbol{z}_e \sim \mathcal{N}(\boldsymbol{\mu}_e, \boldsymbol{\sigma}_e)$. The encoder $E$ corresponds $q_\phi(z|\alpha)$ and the decoder $D$ corresponds $p_\theta(\alpha|z)$. Specifically, the encoder $E$ and the decoder $D$ are implemented by two LSTM networks.

---

**Algorithm 2** VAENAS - One-Shot

---

**Input:** Super-network training iterations $n_{iters}^{sup}$, VAE training iterations $n_{iters}^{vae}$, interval to train VAE $n_{vae}$, # of sampled architectures (rank) $N_s$, # of architectures seed $N_{seeds}$, super-network $G$, search space $\mathcal{A}$, network weights $\mathcal{W}$, exploitation increase parameter $\epsilon_{inc}$, ranking ratio $K$, parameters of VAE module $\phi$ and $\theta$.

**Output:** The final architecture $\alpha^*$

1: Define $\Gamma(\alpha) = (1 - \epsilon) * U(\alpha) + \epsilon * p_\theta(\alpha|z)$; Set $\epsilon = 0$;
2: **for** $i = 1, ..., n_{iters}^{sup}$ **do**
3:     Sample $\alpha \sim \Gamma(\alpha)$;
4:     Update network weights $\mathcal{W}_\alpha$;
5:     **if** $i \mod n_{vae} == 0$ **then**
6:         Sample $N_s$ number of architectures $\{\alpha\}$ from $\Gamma(\alpha)$, $\alpha \sim \Gamma(\alpha)$;
7:         Evaluate $\{\alpha\}$, obtain accuracy $\{ACC\}$;
8:         Set $S = \{(\alpha_i, ACC_i)\}_i^{N_s}$;
9:         Call algorithm 1 VAE$(S, \phi, \theta, n_{iters}^{vae}, K)$ to update VAE module;
10:     **end if**
11:     $\epsilon \leftarrow \epsilon + \epsilon_{inc}$;
12: **end for**
13: Sample $N_{seeds}$ number of architectures $\{\alpha\}$ from $p_\theta(\alpha|z)$, $\alpha \sim p_\theta(\alpha|z)$;
14: Set $S_{seeds} = \{\alpha\}$.
15: Derive the final architecture $\alpha^*$ from $Search(G, \mathcal{A}, S_{seeds})$;

---

# 4 VAENAS FOR NEURAL ARCHITECTURE SEARCH

In this section, we demonstrate how VAENAS is integrated into existing NAS frameworks. Section 4.1 gives the formal definition of weight sharing mechanism. Section 4.2 gives detailed description of VAENAS combined with one-shot framework. Section 4.3 shows VAENAS combined with gradient-based framework.

## 4.1 WEIGHT-SHARING MECHANISM

Weight sharing approaches construct a super-network $G(\boldsymbol{\alpha}, W)$ that contains all the valid architectures $\boldsymbol{\alpha}$ with shared weights $W$. Any architecture $\alpha$ is a sub-graph in the super-network and inherits corresponding weights $W_\alpha$. Thus we can formalize the training of weight sharing mechanism as solving a bi-level optimization problem:

$$W_\alpha^* = \arg\min_{W_\alpha} \mathbb{E}_{\alpha \sim \Gamma(\alpha)}[L_{train}(G(\alpha, W_\alpha))] \tag{3}$$

$$\alpha^* = \arg\min_{\alpha \in \boldsymbol{\alpha}} L_{val}(G(\alpha, W_\alpha^*)) \tag{4}$$

where $\Gamma(\alpha)$ is an architecture distribution. There are two main frameworks which utilize weight-sharing mechanism: one-shot approach and gradient-based approach.

## 4.2 ONE-SHOT FRAMEWORK

**Review of One-Shot Architecture Search.** Generally, one-shot approaches (Liu et al. (2018b); Cai et al. (2018); Xie et al. (2018)) consist of three steps: 1) train a weight sharing super-network. 2) rank architectures based on its performance on the super-network. 3) derive the final architecture via a search strategy. Previous methods typically adopt hand-crafted sampling methods (Bender et al., 2018; Guo et al., 2019; Chu et al., 2019).

**Combine One-Shot Approach with VAENAS.** Algorithm 2 shows the detailed pipeline of VAE-NAS combined with one-shot approach. VAENAS-OS utilizes one-shot framework and optimizes sampling distribution:

$$\Gamma(\boldsymbol{\alpha}) = (1 - \epsilon) * U(\boldsymbol{\alpha}) + \epsilon * p_\theta(\boldsymbol{\alpha}|\boldsymbol{z}) \tag{5}$$

where $U(\boldsymbol{\alpha})$ is random uniform distribution, $p_\theta(\boldsymbol{\alpha}|\boldsymbol{z})$ is variational distribution, and $\epsilon$ is exploitation parameter. On supernet training, architectures are sampled from $p_\theta(\boldsymbol{\alpha}|\boldsymbol{z})$ and $U(\boldsymbol{\alpha})$ with probability

---

**Algorithm 3** VAENAS - Gradient-Based

---

**Input:** Ranking ratio $K$, network training iterations $n_{iters}$, VAE training iterations $n_{iters}^{vae}$, interval to train VAE $n_{vae}$, # of sampled architectures (rank) $N_s$, network weights $\mathcal{W}$, architecture weights $w$, parameters of VAE module $\phi$ and $\theta$.
**Output:** The final architectures $\alpha^*$

1: **for** $i = 1, ..., n_{iters}$ **do**
2:     Sample $\alpha \sim p_\theta(\alpha|z)$;
3:     Update architecture weights $w_\alpha$.
4:     Update super-network weight $\mathcal{W}_\alpha$
5:     **if** $i \mod n_{vae} == 0$ **then**
6:         Sample $N_s$ number of architectures $\{\alpha\}$ from $U(\alpha)$, $\alpha \sim U(\alpha)$;
7:         Evaluate $\{\alpha\}$, obtain accuracy $\{ACC\}$;
8:         set $S = \{(\alpha_i, ACC_i)\}_i^{N_s}$;
9:         Call algorithm 1 VAE$(S, \phi, \theta, n_{iters}^{vae}, K)$ to update VAE module;
10:    **end if**
11: **end for**
12: Derived the final architecture $\alpha^*$ based on the learned $w$.

---

$\epsilon$ and $1 - \epsilon$. As the search phase continues, the $\epsilon$ increases, thus the super-network trains more samples from VAENAS. The good-performing architectures, for example, with $k\%$ highest accuracy, are collected every $n_{vae}$ epochs and fed into the VAENAS module to train. As the training phase proceeds, the variational distribution $p_\theta(\boldsymbol{\alpha}|\boldsymbol{z})$ would converge to generating good-performing architectures. On search phase, the initial seeds are sampled from $p_\theta(\boldsymbol{\alpha}|\boldsymbol{z})$ and then search with genetic algorithm as Guo et al. (2019). Due to the weights of good-performing architectures are trained more adequately, intuitively, the super-network would be more predictive. Algorithm 2 shows the detailed pipeline of VAENAS combined with the one-shot approach.

**Why Architecture-Dependent Sampling for One-Shot?** The conventional sampling methods for one-shot NAS, such as zero out random operations (Bender et al. (2018)), uniform sampling (Guo et al. (2019)) and fair sampling (Chu et al. (2019)) are fixed sampling methods.

One issue of fixed architecture distribution in one-shot approach is its ineffectiveness to sample good-performing architectures, both during super-network training and architecture search. When train a super-network with architectures sampled from a pre-defined architecture distribution such as uniform distribution, it is likely that the best-performing architectures are not picked; thus the search algorithm fails to seek them out due to the low performance of best architectures on the super-network. Moreover, the search strategy usually adopts evolution/RL algorithms, where a set of architectures are selected as seeds via sampling method. Due to the evaluation process is expensive, only a small set of architectures are evaluated and ranked. Thereby the initialized set for search strategy can be sub-optimal. VAENAS resolve above issues by adopting an architecture-dependent distribution on super-network training and search phase. By learning the distribution of good-performing architectures, VAENAS ensures the sampling procedure effective since the samples from the variational distribution $p_\theta(\boldsymbol{\alpha}|\boldsymbol{z})$ are guaranteed to have good performance on the super-network.

### 4.3 GRADIENT-BASED FRAMEWORK

**Review of Gradient-Based Architecture Search.** Gradient-based approaches (Liu et al. (2018b); Cai et al. (2018); Xie et al. (2018)) employ an independent discrete architecture parameter to model the networks, thus we can follow the exact bi-level optimization process as we describe in equation 3 and 4. The common practice of gradient-based approach does not contain sampling process. Proxylessnas (Cai et al. (2018)) introduce a multinomial distribution sampling process prior to the optimization step of architecture weights.

**Combine Gradient-Based Approach With VAENAS.** Alternatively, we consider replacing the sampling procedure by VAENAS:

$$\Gamma(\boldsymbol{\alpha}) = p_\theta(\boldsymbol{\alpha}|\boldsymbol{z}) \tag{6}$$

Table 1: Comparison with state-of-the-art architectures on CIFAR-10 under the NASNET-like search space. * denotes model is trained with cutout. OS denotes one-shot approach. G denotes gradient-based approach. † indicates hard parameters constraint in search phase.

| Architecture | Test Error (%) | Params (M) | Search Method |
|---|---|---|---|
| NASNet-A[*] (Zoph et al. (2018)) | 2.65 | 3.3 | RL |
| AmoebaNet-B[*] (Real et al. (2019)) | 2.55±0.05 | 2.8 | Evolution |
| Hierarchical Evolution (Liu et al. (2017)) | 3.75±0.12 | 15.7 | Evolution |
| PNAS (Liu et al. (2018a)) | 3.41±0.09 | 3.2 | SMBO |
| ENAS[*] (Pham et al. (2018)) | 2.89 | 4.6 | RL |
| NAO-weight-sharing [*](Luo et al. (2018)) | 3.53 | 2.5 | Gradient |
| DARTS[*] (Liu et al. (2018b)) | 2.76±0.09 | 3.4 | Gradient |
| SNAS[*] (Xie et al. (2018)) | 2.76±0.09 | 3.4 | Gradient |
| GraphHypernet[*] (Zhang et al. (2018a)) | 4.3±0.1 | 5.1±0.6 | Gradient |
| DSO-share[*] (Zhang et al. (2018b)) | 2.84±0.07 | 3.0 | Gradient |
| BayesNAS[*] (Zhou et al. (2019)) + $\lambda = 0.01$ | 2.81±0.04 | 3.4 | Gradient |
| VAENAS-G[*] | 2.40 | 4.4 | Gradient |
| VAENAS-OS[*]† | 2.50 | 3.4 | Evolution |
| VAENAS-OS[*] | 2.26 | 5.2 | Evolution |

Specifically, we sample an architecture $\alpha_i$ from $p_\theta(\alpha|z)$, and update super-network weights $W_{\alpha_i}$ and multinomial distribution parameters $x_i$ accordingly. Algorithm 3 shows the detailed pipeline.

**Why Independent Sampling Module Necessary?** Conventional gradient-based methods (Liu et al. (2018b)) do not contain the sampling process: all architecture operations are updated during training at each iteration. These methods bring overwhelming memory and computation consumption issue, such that direct model search and direct dataset search are incapable. Proxylessnas (Cai et al. (2018)) present multinomial distribution to sample the architectures in the search phase.

However, the probability of multinomial distribution is obtained by applying softmax to architecture weights. The principal limitation is that the model weights and architecture weights are mutually interactional. To be concrete, if an architecture gets more training, more likely it will be sampled again in the next iteration. This *winner takes all* effect causes premature convergence and reduces the diversity of architectures. Moreover, the gradient tends to flow through non-parameter operations, such as identity (the architectures derived by gradient-based method is typically small, i.e., 3.4M on CIFAR10 (Liu et al. (2018b)). The VAENAS sampling module, which serves as an independent sampling module, helps to 1) increase the diversity of architectures that can be exploited in the training process, as a result, alleviate the premature convergence phenomenon, 2) search for large models. Large models naturally obtain better performance than small models.

## 5 EXPERIMENTS

### 5.1 NASNET-LIKE SEARCH SPACE.

The detailed training strategy, search space and hyper-parameters settings can be found in Appendix A.1. Our VAENAS is evaluated on both gradient-based and one-shot approaches, and compared with the state-of-the-art NAS methods under the same search space. We use 'OS' to denote one-shot approach and use 'G' to denote gradient-based approach.

**Performance on CIFAR-10.** We present the results on Table 1. Combined with gradient-based method, our method achieves 2.4% test error with 4.4M parameters. Combined with one-shot approach, our method gives two results. The first one is searched with parameter constraint of 3.5M parameters, and obtain 2.5% test error, and the second one is trained without any constraint and obtain 2.26% test error with 5.2M parameters.

**Performance on ImageNet.** We present the results on Table 2. Combined with gradient-based approach, our method obtains 75.6% top-1 accuracy with 568M FLOPs. Combined with one-shot

Table 2: Comparison with state-of-the-art architectures on ImageNet (mobile setting) under NASNET-like search space. † denotes direct search on ImageNet.

| Architecture | FLOPs (M) | Params (M) | Top-1 Acc. (%) | Top-5 Acc. (%) |
|---|---|---|---|---|
| NASNet-A (Zoph et al. (2018)) | 564 | 5.3 | 74.0 | 91.6 |
| AmoebaNet-A (Real et al. (2019)) | 555 | 5.3 | 74.0 | 91.5 |
| AmoebaNet-B (Real et al. (2019)) | 555 | 5.1 | 74.5 | 92.0 |
| AmoebaNet-C (Real et al. (2019)) | 570 | 6.4 | 75.7 | 92.4 |
| PNAS (Cai et al. (2018)) | 588 | 5.1 | 74.2 | 91.9 |
| GraphHypernet (Zhang et al. (2018a)) | 569 | 6.1 | 73.0 | 91.3 |
| BayesNAS (Zhou et al. (2019)) | - | 3.9 | 73.5 | 91.1 |
| DARTS (Liu et al. (2018b)) | 574 | 4.7 | 73.3 | 91.3 |
| SNAS (Xie et al. (2018)) | 522 | 4.3 | 72.7 | 90.8 |
| VAENAS-G† | 568 | 6.0 | 75.6 | 92.3 |
| VAENAS-OS† | 573 | 6.1 | 75.8 | 92.7 |

Table 3: Comparison with state-of-the-art architectures on ImageNet (200M-400M FLOPs) under ShuffleNet-like search space. * denotes results reported using AutoAugment.

| Architecture | FLOPs (M) | Params (M) | Top-1 Acc. (%) | Top-5 Acc. (%) |
|---|---|---|---|---|
| ShuffleNet V2 1.5 (Ma et al. (2018)) | 300 | - | 72.6 | - |
| MobileNet V2 1.0 (Sandler et al. (2018)) | 300 | 3.4 | 72.0 | 91.0 |
| MobileNet V3 Large 1.0 (Howard et al. (2019)) | 219 | 5.4 | 75.2 | 92.2 |
| MnasNet-A2 (Tan et al. (2019)) | 340 | 4.8 | 75.6 | 92.7 |
| FBNet-B (Wu et al. (2019)) | 295 | 4.5 | 74.1 | - |
| Proxyless GPU (Cai et al. (2018)) | 320 | 4.0 | 74.6 | 92.2 |
| Single-Path NAS (Guo et al. (2019)) | 365 | 4.3 | 75.0 | 92.2 |
| FairNAS-A (Chu et al. (2019)) | 388 | 4.6 | 75.3 | 92.4 |
| EfficientNet-B0* (Tan & Le (2019)) | 390 | 5.3 | 76.3 | 93.2 |
| Baseline | 360 | 6.7 | 77.1 | 93.3 |
| VAENAS-OS | 365 | 6.7 | 77.4 | 93.6 |

approach, our method obtains 75.8% top-1 accuracy with 573M FLOPs. Both results surpass previous state-of-the-art results.

## 5.2 ImageNet Classification under Mobile Setting

**Comparison with State-of-the-art methods.** Table 3 lists a number of state-of-the-art models' FLOPs on ImageNet dataset. We select models with the range of FLOPS from 200M to 400M. Notably, VAENAS-OS outperforms state-of-the-art models by a large margin. In particular, the accuarcy of VAENAS is 1.1% higher than EfficientNet-B0 (Tan & Le, 2019), recent state-of-the-art method, while using 6.5% less computational budget. To best of our knowledge, VAENAS-OS is the first model surpass 77% under 400M FLOPs without using strong regularization and data augmentation techniques such as AutoAugment.

## 5.3 VAENAS on NAS-Bench-101 Dataset

NAS-Bench-101 dataset (Ying et al., 2019) is a carefully constructed architecture dataset for NAS research, including over 5 million already trained architectures on CIFAR-10. The ground-truth accuracy of architectures can be gotten from NAS-Bench-101 dataset directly. We compare VAENAS sampling with uniform sampling (also known as random search Li & Talwalkar (2019)).

Table 4: Comparison of VAENAS and random search on NAS-Bench-101 dataset.

| Methods | Max accuracy for K samples (%) | | | | Evolution |
|---|---|---|---|---|---|
| | 10 | 50 | 100 | 300 | |
| Random Search | 92.93 | 93.30 | 93.53 | 93.86 | 93.39 |
| Weight-sharing | 93.72 | 93.95 | 93.98 | 94.23 | 94.12 |
| Non-weight-sharing + noise 0 | 93.56 | 93.71 | 93.95 | 94.13 | 94.14 |
| Non-weight-sharing + noise 0.1% | 93.64 | 93.73 | 93.97 | 94.17 | 93.75 |
| Non-weight-sharing + noise 0.5% | 92.24 | 93.35 | 93.63 | 93.83 | 93.33 |

We design two methods for training VAENAS. The weight-sharing method follows Section 4.1, where VAENAS is trained with a super-network on NAS-Bench-101 search space. The design of super-network is described in Appendix A.5. The non-weight-sharing method trains VAENAS with a set of high-accuracy architectures. 6000 trained architectures are sampled and the top-1500 models form the training set.

**Influence of noise on accuracy.** The training of VAENAS is based on high-accuracy architectures, while ground-truth accuracy of architectures is hard to be obtained in most cases. We add various levels of noise on the accuracy, then rank architectures and select high-accuracy architectures to train corresponding VAENAS module. As displayed in Table 4 and Fig. 2, VAENAS is robust to a certain extent of noise. Futhermore, the accuracy of architectures in weight-sharing methods have a noise on super-network, while the accuracy on super-network is inconsistent with ground-truth accuracy. The results of VAENAS trained by weight-sharing method proof that VAENAS works well with NAS frameworks.

**Comparisons of sampling methods.** We sample a number architectures by VAENAS and random method and evaluate the maximum accuracy of architectures, which is shown in Table 4. VAENAS, trained with super-network or clear ground-truth accuracy, outpeforms random sampling (+0.79% with 10 samples). In addition, we sample 5000 architectures with random sampling and VAENAS sampling, then display the distribution of samples in Fig. 2. The distribution of VAENAS covers high-accuracy architectures, while the distribution of random sampling is relatively dispersed.

**Combination with evolution search.** We execute evolution search on NAS-Bench-101 and replace random sampling by VAENAS during the initialization of population. The results are shown in Table 4. VAENAS outperforms uniform sampling by 0.7%. It proves that VAENAS is helpful to the search strategy, since the convergence of evolution search is accelerated.

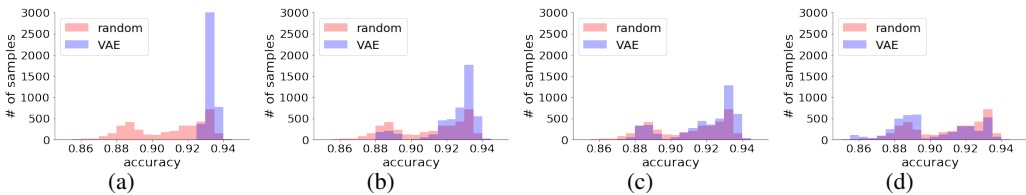

Figure 2: Compare the distribution of VAE Sampling with Random Sampling. (a) VAENAS trained with super-network. (b) VAENAS trained with ground truth accuracy. (c) VAENAS trained with 0.1% noise on accuracy. (d) VAENAS trained with 0.5% noise on accuracy.

## 6    CONCLUSION

In this paper, we argue sampling matters in NAS and present a learnable sampling method based on VAE. The VAENAS learns the architecture distribution and is interactive with NAS search process. As a result, by embedding into the existing efficient NAS framework, we achieve a series of state-of-the-art results on various dataset and search space. Overall, we believe that the VAENAS approach provides a practical new perspective on sampling methods in neural architecture search.

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

Table 5: The comparison of architectures searched with (w/) or without (w/o) VAENAS.

| Model | Dataset | | w/o VAENAS | | w/ VAENAS | |
|---|---|---|---|---|---|---|
| | CIFAR-10 | Params | Top-1 Acc. | Params | Top-1 Acc.(gain.) |
| Gradient | ✓ | 3.7M | 2.84% | 4.4M | 2.40%(+0.44%) |
| One-Shot | ✓ | 4.3M | 3.33% | 5.2M | 2.26%(+0.93%) |
| Model | Dataset | | w/o VAENAS | | w/ VAENAS | |
| | ImageNet | FLOPs | Top-1 Acc. | FLOPs | Top-1 Acc.(gain.) |
| Gradient | ✓ | 519M | 74.1% | 568M | 75.6%(+1.5%) |
| One-Shot | ✓ | 553M | 73.3% | 573M | 75.8%(+2.5%) |

## A    APPENDIX

### A.1    IMPLEMENTATION DETAILS OF VAENAS

**VAENAS-OS ShuffleNet training details.** VAENAS-OS is trained for 480 epochs with batch size 1024 on 8 2080-Ti GPUs. It is based on the Pytorch (Paszke et al. (2017)). The network parameters are optimized using an SGD optimizer with an initial learning rate of 0.5 (decayed linearly after each iteration), a momentum of 0.9 and a weight decay of $3 \times 10^{-5}$. Additional enhancements including label smoothing (Szegedy et al. (2016)) and Dropout with 0.5 on last FC layer. We follow the common practice to use inception augmentation. No other data augmentation method is used.

*VAENAS-OS/VAENAS-G NASNet training details:*
**ImageNet.** The model is trained for 300 epochs with batch size 512 on 8 2080-Ti GPUs. It is based on the Pytorch (Paszke et al. (2017)). The network parameters are optimized by using an SGD optimizer with an initial learning rate of 0.5 (decayed linearly after each iteration), a momentum of 0.9 and a weight decay of $3 \times 10^{-5}$. Additional enhancements include label smoothing (Szegedy et al. (2016)) and auxiliary tower with 0.4. We follow the common practice to use inception augmentation. The initial width is 48 and depth is 14, which follows the same setting as Liu et al. (2018b).

**CIFAR-10.** The model is trained by following the common pipeline (Liu et al. (2018b); Real et al. (2019); Zoph et al. (2018); Pham et al. (2018); Liu et al. (2018a)). it is trained for 600 epochs with batch size 48 on single GPU. Additional enhancements include label smoothing (Szegedy et al. (2016)), auxiliary tower with 0.4, drop path with 0.2 and Cutout (DeVries & Taylor (2017)).

### A.2    IMPORTANCE OF VAENAS

More ablation study will be available in the Appendix A.3. In this section, we use gradient-based NAS approach and one-shot NAS approach as two baseline framework, and compare the result with VAENAS and without VAENAS using the same search and training strategy. The baseline method of gradient-based approach is re-implemented based on Cai et al. (2018) and one-shot approach is re-implemented based on Guo et al. (2019). The results are shown in Table 5. We can observed that the the framework with VAENAS achieve significantly better performance on both CIFAR-10 and ImageNet.

### A.3    ABLATION STUDY

**Influence of Depth Gap between Search and Evaluation.** One issue encountered in prior works with both one-shot NAS approach and gradient-based approach is the challenge of search on the network with the same depth as the evaluated models. We therefore study the effect of VAENAS from the perspective of direct model search. Table 6 shows the results of baseline with different depth that one-shot method fails to find the good architecture when directly search on target model. One explanation is that the search space is too large for the conventional uniform sampling.

Table 6: NAS framework with different model depth on CIFAR-10.

| Model | Params | Top-1 Acc. | Search Depth |
|---|---|---|---|
| One-Shot Baseline | 4.3M | 3.33% | 20 |
| One-Shot Baseline | 3.7M | 2.89% | 8 |
| VAENAS-OS | 5.2M | 2.26% | 20 |
| One-Shot Baseline | 3.4M | 2.91% | 8 |
| One-Shot Baseline | 3.7M | 2.84% | 20 |
| VAENAS-Gradient | 4.3M | 2.40% | 20 |

## A.4 NETWORK ARCHITECTURE

This section shows the details of network structures which we searched for NASNet-like search space and ShuffleNet-like search space.

**ShuffleNet-like search space.** We designed a search space based on ShuffleNetV2. Specifically, we construct a over-parameterized network with 20 layers, each layer can choose between four operations:

- Shuffle Convolution 3x3
- Shuffle Convolution 5x5
- Shuffle Convolution 7x7
- Shuffle Xception 3x3

ReLU non-linearity are replaced by Swish if conv 5x5, conv 7x7 or Xception are chosen. Squeeze-and-Exited module are also adopted if conv 7x7 and Xception are chosen. We utilized VAENAS combined with one-shot framework to do the search.

**NASNet search space.** We modify the search space of DARTS (Liu et al., 2018b). A cell is a directed acyclic graph, and every edge in the graph represents a kind of computation operation. Different from DARTS, in our search space, one node could have more than two predecessors in one cell. The maximum number of edges in one cell is $2 + 3 + 4 + 5 = 14$, while 'None' is an available operation in our search space. The list of candidate operations is listed as follows:

- None
- Average Pooling 3x3
- Max Pooling 3x3
- Skip Connect
- Separate Convolution 3x3
- Separate Convolution 5x5
- Separate Convolution 7x7
- Dilated Convolution 3x3

The magnitude of search space is $8^{2 \times 14}$, while normal cell and reduction cell are searched jointly. We present the cell searched by our methods in Fig.4 and Fig.5.

## A.5 SUPER-NETWORK AND VAENAS ON NAS-BENCH-101 SEARCH SPACE

**Super-network.** NAS-Bench-101 is a cell-based search space, and the details of NAS-Bench-101 search space is introduced in Ying et al. (2019). While NAS-Bench-101 is not proposed for weight-sharing NAS methods, we design one super-network which covers architectures in NAS-Bench-101.

The output of one cell in NAS-Bench-101 search space is the concatenation of input nodes. The illustration Fig.6 shows one sampled cell on the super-network. However, the output channels of

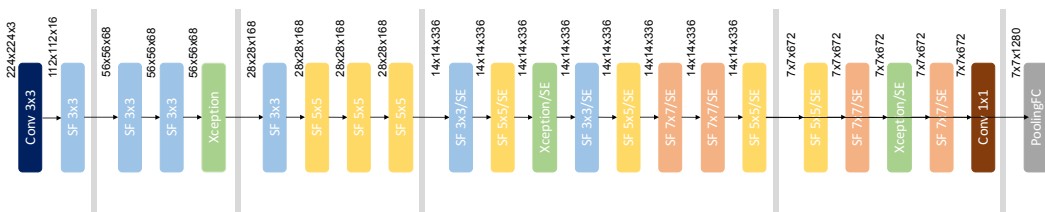

Figure 3: VAENAS-OS ShuffleNet architectures. We highlight the input and output tensor shape. Conv denotes convolution layer. SF denotes ShuffleNetV2 module. SE denotes Squeeze-and-Excitation module.

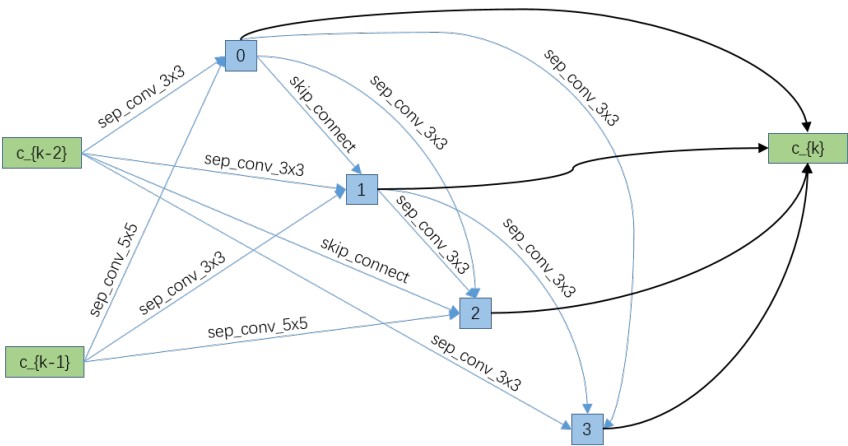

Figure 4: VAENAS-OS normal cell trained on CIFAR-10.

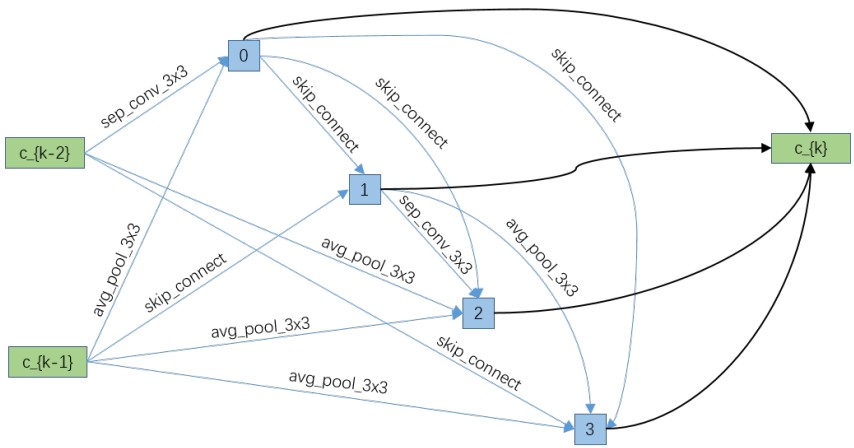

Figure 5: VAENAS-OS reduction cell trained on CIFAR-10.

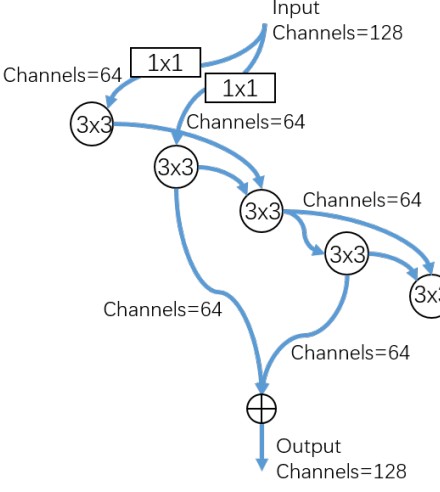

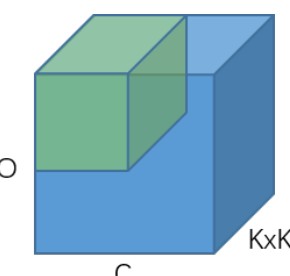

Figure 6: One sampled cell on NAS-Bench-101 super-network. Every path contains 9 edges, while some edges may be useless in the cell.

Figure 7: One example of weight-shared tensors for convolution kernel. $O \times C \times K \times K$ is the size of kernel. The blue cube shows the whole tensor, and the green cube shows the part used for the $\frac{O}{2} \times \frac{C}{2} \times K \times K$ convolution operation.

one middle node is not fixed, so weight-shared tensor is utilized to save the parameters of convolutional operations. The illustration Fig.7 shows one example of dividing convolutional kernel for computation.

**Representation and Sampling.** In NAS-Bench-101 dataset, one architecture can be represented as a 7x7 upper-triangle adjacent matrix and 5 operations. To simply the representation of architectures, the adjacent matrix is converted to a list of edges $\boldsymbol{a} = \{a_1, a_2, ..., a_{21}\}, a_i \in \{0, 1\}$, and operations compose a vector $\boldsymbol{op} = \{op_1, op_2, ..., op_5\}, op_i \in \{conv1\times1, conv3\times3, maxpool\}$. The integrated architecture is a vector $\alpha = concat(\boldsymbol{a}, \boldsymbol{op})$.

Two constraints of edges determine the validity of one sampled cell. One is the number of edges in one cell is no more than 9 in a cell. The other is that at least one path connect the input node and output node in a cell, which indicate the connectivity of the cell. Under these two constraints, we design the following sampling strategy. 21 number in range $[0, 1]$ is sampled by random sampling or VAE sampling, then the indexes of top-9 numbers is selected as $IDX$. We set $a_i = 1$ when $a_i \in IDX$, and set $a_j = 0$ when $j \notin IDX$. If the sampled cell is not connected, we perform above sampling process again.

**Evolution Search on NAS-Bench-101.** The setting of evolution search follows (Guo et al. (2019)). In addition, to make sure the validity of descendants, the mutation and crossover of the edge list $a$ should be planned carefully.

For mutation, an index $i, a_i = 1$ and another index $j, a_j = 0$ are selected randomly. Another new sample $a'$ is generated by exchanging $a_i$ and $a_j$. We repeat this process until $a'$ satisfy connectivity.

For crossover, two edge lists $a^x$ and $a^y$ are selected randomly. Then an index $i, a_i^x = 1, a_i^y = 0$ and another index $j, a_j^x = 0, a_j^y = 1$ are selected. One new sample $a'$ is generated by exchanging $a_i^x$ and $a_j^x$. If $a'$ does not maintain connectivity, it will be dropped, and the crossover process will be repeated.

