# OpenReview forum: "VAENAS: Sampling Matters in Neural Architecture Search"
_ICLR.cc/2020/Conference — Reject_

### Official Review · AnonReviewer1 · 2019-10-22
**Official Blind Review #1**

**Rating:** 3

**Review:**

============ comments after rebuttal
I would like to thank the authors for addressing some of my concerns. I believe the new results under Q2 and Q3 are useful additions to strengthen the paper.

As for the authors' comments for Q1, I'd like to point out that a "larger" search space is not necessarily more difficult (a more meaningful metric would be the average accuracy of random architectures). It is still possible that the current search space is putting the proposed method at advantage, especially if having dense connections is a useful prior.

Given the above, I would like to increase my score from 1 to 3 (weak reject).

============ previous comments
Neural architecture search can be formulated as learning a distribution of promising architectures (the sampling policy). Such a distribution is usually represented in a fully factorized fashion (e.g., as a set of multinational distributions as in DARTS). This paper proposes to model the architecture distribution using a VAE instead, where the encoder and decoder are implemented using LSTMs. The authors argue that the increased flexibility of the sampling policy leads to improved performance on CIFAR-10, NASBench and ImageNet.

The idea of representing the architecture distribution using VAEs is very natural, which in principle could offer better coverage over interesting regions in the search space as compared to traditional factorized distribution representation (which has a single mode only).

While the method itself is interesting, I do not think it has been properly backed up by controlled experiments. This is largely due to the fact that the authors are comparing their method against baselines in fundamentally different search spaces. For instance:
* For CIFAR-10 experiments, the authors mentioned in the appendix: "Different from DARTS, in our search space, one node could have more than two predecessors in one cell". This makes the search space very different from the existing ones as used by NASNet/AmoebaNet/DARTS/SNAS, and it hence remains unclear to what degree the resulting architecture has benefited from the increased in-degrees per node. Note the searched densely connected cells in Figure 4 & 5 in Appendix A.4 are clearly not part of the search space for many of the baselines.
* For ImageNet experiments, the authors are using a ShuffleNet-like search space which has fundamentally different building blocks than other architecture search baselines (commonly built on top of inverted bottleneck layers). It is unclear to what degree the 77.4 top-1 accuracy @ 365 MFlops results have benefited from this different search space.

Without fair comparisons in a controlled setup, it is impossible for readers to draw any solid conclusion about the true empirical advantages of the method. I'm therefore unable to recommend acceptance for the paper at the moment, but am willing to raise my score if the authors can properly address those issues in the rebuttal.

Additional question: How can we isolate it to tell whether the gains come from the LSTMs or the VAEs? Is there any intuition why incorporating a generative sampler based on VAEs is potentially superior to method like ENAS (which involves LSTMs decoders only)?

**Experience Assessment:**

I have published one or two papers in this area.

**Review Assessment: Checking Correctness Of Derivations And Theory:**

I carefully checked the derivations and theory.

**Review Assessment: Checking Correctness Of Experiments:**

I carefully checked the experiments.

**Review Assessment: Thoroughness In Paper Reading:**

I read the paper thoroughly.

---

> ### Author Response · Authors · 2019-11-15
> **Author Response to Official Blind Review #1**
>
> Thank you very much for your constructive comments.
>
> Q1: For CIFAR-10 experiments, the authors mentioned in the appendix... Note the searched densely connected cells in Figures 4 & 5 in Appendix A.4 are clearly not part of the search space for many of the baselines.'
> A1: We admit that the search space in our experiments is different from DARTS, as we mentioned in Appendix A.4. However, it is necessary to note the difference between the search space that we used in our paper and the DARTS search space.  The difference between our search space and darts search space is that: in our search space, the i-th node of one cell has i-1 processors, while in DARTS search space, the i-th node of one cell has 2 processors. However, in our search space, 'None' is an available operation. Choosing 'None' means delete the corresponding edge in the cell. Thus, the DARTS search space is a ***subset*** of ours. In other words, our search space is much larger than the search space used in DARTS paper. The reason we adopt this search space it that, it is easier to verify the effectiveness of a sampling module on a larger search space.
>
> However, we understand your concern about the unfair comparison. Due to the time limit, we only finish some experiments with the DARTS search space on CIFAR-10 and we report the following results:
> 97.46% with 2.92M parameters, searched by one-shot based VAENAS.
> The preliminary experiments show that our VAE sampling is still effective in the DARTS search space. We are doing more experiments and we will update the results in a future revision.
>
> Q2: Unfair comparison of using ShuffleNet search space for ImageNet experiments.
> A2: For the ShuffleNet-like search space, we claim that our contribution is to find an architecture to achieve SOTA results on ImageNet under 400M FLOPs. Regarding the fairness, we include the baseline (results that are searched by one-shot) in the new revision. The baseline is 77.1% with 360M FLOPs, compare to 77.4% with 365M FLOPs with our methods.
>
> Q3: What's the gains come from the LSTMs or the VAEs?
> A3: Regarding the concern on performance gain from LSTM, we did experiments on VAE use MLPs (Multi-Layer Perception) as encoder and decoder. In our search space, we got 97.60% with 4.6M on the CIFAR-10 dataset. The preliminary experiments indicate that the performance gain from LSTM is limited.
> Secondly, we do not claim that our generative sampler based on VAEs is superior to other samplers. Our main contribution is that we design a sampling module based on VAE that can supplementary to the existing weight sharing framework (one-shot and gradient-based). We use VAE because it is lightweight and easy to train.

---

### Official Review · AnonReviewer2 · 2019-10-25
**Official Blind Review #2**

**Rating:** 3

**Review:**

This paper proposed to use VAE to learn a sampling strategy in neural architecture search. The main idea is to use the currently high-performing networks to train a VAE from which the sampled architectures for the next iteration will likely supply both high-performing networks and better diversity coverage. The experiments are extensive, including results under various settings.

The idea is straightforward and reasonable. I do not work on neural architecture search myself, so I'm not sure how significant the experimental results are. Would the 0.1% (1.1%) absolute improvement over the second best in Table 2 (Table 3) be considered significant enough to justify the effectiveness of the proposed approach?

I'm a little concerned about the fairness of the comparison experiments. A fairly heavy computation overhead is required to train the VAE models in the proposed method. Instead of taking this overhead, wouldn't it be easier to randomly sample more architectures? Intuitively, if we spend the cost of training a VAE model instead on sampling more architectures, the end  effects could be the same.

Are the numbers in Table 2 and Table 3 swapped?



**Experience Assessment:**

I have read many papers in this area.

**Review Assessment: Checking Correctness Of Derivations And Theory:**

I carefully checked the derivations and theory.

**Review Assessment: Checking Correctness Of Experiments:**

I carefully checked the experiments.

**Review Assessment: Thoroughness In Paper Reading:**

I read the paper thoroughly.

---

> ### Author Response · Authors · 2019-11-15
> **Author Response to Official Blind Review #2**
>
> Thank you very much for your constructive comments.
>
> Q1:Would the 0.1%  absolute improvement over the second best in Table 2  be considered significant enough to justify the effectiveness of the proposed approach? '
> A1: The 0.1% improvement is not significant. However, it is a comparison between VAENAS-OS with AmoebaNet-C, which is searched with a computation cost of 3150 GPU days. While the search cost of recent NAS methods is less than 1 GPU day, including our method, it is unfair to make a comparison between our methods with AmoebaNet-C.
>
> Q2: Would the 1.1% absolute improvement over the second best in Table 3 be considered significant enough to justify the effectiveness of the proposed approach? '
> A2: The 1.1% accuracy improvement in Table 2 is significant. Improving the accuracy of ImageNet classification is hard, especially when it comes to efficient models (< 600M FLOPs).
>
> Notably, we present a controlled experiment to demonstrate how effective is VAE sampling module, and the results are shown in Table 5.  The results on both CIFAR10 and ImageNet with gradient-based/one-shot is significant. We think this can justify that our sampling module is effective.
>
> Q3: Training budget and fairness regarding random sample?
>
> Regarding the fairness of the comparison experiments, we decouple the training time (computation overhead) into VAE training stage and sub-networks evaluation stage.
>
> We argue that the computation overhead of VAE sampling module itself is lightweight. Our VAE module was composed of one LSTM as an encoder, one LSTM as a decoder, and several FC layers, so the parameter size of VAE module is less than 50 KB. To be specific, we train super-network for 100 epochs, and we train VAE for 50 epochs every 10 super-network epochs. According to our test, training VAE with 50 epochs takes 0.3% training time to train a one-shot super-network with one epoch. It is also equivalent to evaluate 3-5 architectures on a super-network.
>
> Admittedly, we need to evaluate a number of sub-network to train our VAE module. In our experiment setting, we sample 1,500 sub-networks from the search space and do the evaluation on super-network accordingly. Since we train VAE every 10 epochs, there will be a total of 15,000 sub-networks we evaluated during searching.
>
> However, using the same number of evaluated architectures, random sampling has no way to be more effective than VAE sampling. Our motivation for this paper, and neural architecture search in general, is that random sampling is extremely inefficient in a large search space.
>
> Our experiment in Section 5.3 also shows that to get the same performance architectures, random search has to sample much more than VAE. For example, as shown in Table 4, to get one 93.86% accuracy architecture on the NAS-Bench-101 benchmark dataset, random search sampled 500 architectures but VAE module sampled less than 50 samples.
>
> Q4: Are the numbers in Table 2 and Table 3 swapped? ‘
> A4:  We checked Table 2 and Table 3 and the numbers in Table 2 and Table 3 are correct. We also updated a new version.

---

### Official Review · AnonReviewer3 · 2019-10-28
**Official Blind Review #3**

**Rating:** 3

**Review:**

This paper proposes to use the variational auto-encoder (VAE) to sample the network architectures. The VAE is applied to both one-short and gradient descent scenarios, and shows consistent improvement on different NAS tasks. The proposed method is reasonable, but I have two major concerns:

- I wonder whether the VAE based approach will consistently converge to a good local minimum. It will be very helpful if the authors could provide robust analysis or at least the variations of testing errors.

- I understand the motivation of VAE + one shot, but I am not very convinced by VAE + gradient-based. In the last paragraph in Section 4, the paper claims (1) VAENAS-G can increase the diversity of architectures, which can be also achieved by sampling the data set. Also, it claims (2) VAENAS-G helps to search for large models, which  I do not see experimental supports.

Detailed comments:
- Algorithm 1: it is confusing to use S_K and S_k without explanation
- Table 1: most of the other methods provide test error with standard variations. To be fair, I'd see VAENAS's test error variation.
- Tables 2 and 3: Why is VAENASNet (table 3) different from VAENAS-G and VAENAS-OS?


**Experience Assessment:**

I have read many papers in this area.

**Review Assessment: Checking Correctness Of Derivations And Theory:**

I assessed the sensibility of the derivations and theory.

**Review Assessment: Checking Correctness Of Experiments:**

I assessed the sensibility of the experiments.

**Review Assessment: Thoroughness In Paper Reading:**

I read the paper thoroughly.

---

> ### Author Response · Authors · 2019-11-15
> **Author Response to Official Blind Review #3**
>
> Thank you very much for your constructive comments.
>
> ' I wonder whether the VAE based approach will consistently converge to a good local minimum……at least the variations of testing errors.’
>
> Q1: Whether the VAE based approach will consistently converge to a good local minimum?
> A1: We performed some experiments to verify that if the VAE sampling based approach can consistently find good-performing architectures. Due to limit time constraints, we run four independent searches by VAENAS-OS on CIFAR-10. The average accuracy is 97.56% and the standard deviation is 0.122%. This result is robust in our view.
>
>
> Q2: Why VAENAS-G can increase the diversity of architectures in gradient-based NAS methods?
> A2: We observed that gradient-based methods have the premature convergence problem, which was also observed by concurrent papers [1, 2]. We conjecture that premature convergence phenomenon happens because gradient-based methods have the tendency to converge to the architecture that has better performance at the early search phase, and then sample this architecture repeatedly at the search phase. As a result, only a small number of architectures are actually trained by gradient-based NAS.  Therefore, we claim that our method can increase the diversity of architectures trained by gradient-based methods, such that these NAS methods can make better decisions and search for architectures with higher performance. By "increase the diversity", we mean the VAE sampling module helps the gradient-based method to train with more diversified architectures.
>
> Q3: Why VAENAS-G helps to search for larger models in gradient-based NAS methods?
> A3: We observed that the gradient-based method could be stuck on identity operation [3], result in finding small architectures (with many identity operations). With the help of the sampling module, the gradient-based method can potentially ignore the identity operation because other operations like 3x3 convolution general have better performance than identity. However, the original gradient-based method fails to find large networks because they don't have the opportunity to train them in the first place.
> We presented the experiment results in Table 5 to show that, with the help of VAE sampling module, the gradient-based methods are able to find larger architectures (more parameters or larger FLOPs).
>
> Q4: Detailed comments on notations and tables.
> A4: Thank you for pointing out our mistakes. We fixed these errors in the new revision. We will provide the error variation in future revision.
>
>
>
> [1] BETANAS: Balanced Training and Selective Drop for Neural Architecture Search. https://openreview.net/forum?id=HyeEIyBtvr
> [2] Stabilizing DARTS with Amended Gradient Estimation on Architecture Parameters. https://openreview.net/forum?id=BJlgt2EYwr
> [3] Chen Xin, et al. Progressive Differentiable Architecture Search: Bridging the Depth Gap between Search and Evaluation. In ICCV 2019

---

### Decision · Program_Chairs · 2019-12-19

**Decision:**

Reject

**Comment:**

This paper proposes to represent the distribution w.r.t. which neural architecture search (NAS) samples architectures through a variational autoencoder, rather than through a fully factorized distribution (as previous work did).

In the discussion, a few things improved (causing one reviewer to increase his/her score from 1 to 3), but it became clear that the empirical evaluation has issues, with a different search space being used for the method than for the baselines. There was unanimous agreement for rejection. I agree with this judgement and thus recommend rejection.